# Bmi-1 Immunohistochemical Expression in Endometrial Carcinoma is Correlated with Prognostic Activity

**DOI:** 10.3390/medicina56020072

**Published:** 2020-02-12

**Authors:** Kayo Horie, Chihiro Iseki, Moe Kikuchi, Keita Miyakawa, Mao Yoshizaki, Haruhiko Yoshioka, Jun Watanabe

**Affiliations:** 1Department of Bioscience and Laboratory Medicine, Hirosaki University Graduate School of Health Sciences, Hirosaki, Aomori 036-8564, Japan; yoshioka@hirosaki-u.ac.jp (H.Y.); watajun@hirosaki-u.ac.jp (J.W.); 2Department of Medical Technology, Hirosaki University School of Health Sciences, Hirosaki, Aomori 036-8564, Japan; ci.hydrangea@gmail.com (C.I.); moemoekikuchi@gmail.com (M.K.); miyakawa@asahikawa-med.ac.jp (K.M.); mika-love929@outlook.jp (M.Y.)

**Keywords:** Bmi-1, endometrial carcinoma, immunohistochemical staining, biomarker

## Abstract

*Background and objectives:* B-lymphoma Mo-MLV insertion region 1 (Bmi-1) is a stem cell factor that is overexpressed in various human cancer tissues. It has been implicated in cancer cell proliferation, cell invasion, distant metastasis, and chemosensitivity, and is associated with patient survival. Several reports have also identified Bmi-1 protein overexpression in endometrial carcinoma; however, the relationship between Bmi-1 expression and its significance as a clinicopathological parameter is still insufficiently understood. Accordingly, the present study aimed to clarify whether immunohistochemical staining for Bmi-1 in human endometrial carcinoma and normal endometrial tissues can be used as a prognostic and cell proliferation marker. *Materials and Methods:* Bmi-1 expression was assessed in endometrioid carcinoma (grade 1–3) and normal endometrial tissues (in the proliferative and secretory phases) by immunohistochemistry; protein expression was evaluated using the nuclear labeling index (%) in the hot spot. Furthermore, we examined other independent prognostic and proliferation markers, including the protein levels of Ki-67, p53, and cyclin A utilizing semi-serial sections of endometrial carcinoma tissues. *Results:* The expression of the Bmi-1 protein was significantly higher in all grades of endometrial carcinoma than in the secretory phase of normal tissues. Moreover, Bmi-1 levels tended to be higher in G2 and G3 tissues than in G1 tissue, without reaching significance. Bmi-1 expression showed no notable differences among International Federation of Gynecology and Obstetrics (FIGO) stages in endometrial carcinoma. Furthermore, we observed a significant positive relationship between Bmi-1 and Ki-67, cyclin A, or p53 by Spearman’s rank correlation test, implying that high Bmi-1 expression can be an independent prognostic marker in endometrial carcinoma. *Conclusions:* Our study suggests that Bmi-1 levels in endometrial carcinoma tissues may be useful as a reliable proliferation and prognostic biomarker. Recently, the promise of anti-Bmi-1 strategies for the treatment of endometrial carcinoma has been detected. Our results provide fundamental data regarding this anti-Bmi-1 strategy.

## 1. Introduction

Endometrial carcinoma is the most prevalent gynecological tumor in developed countries. In Japan, its incidence has progressively increased in recent years in association with changes in lifestyle [1]. Endometrial carcinoma is most commonly diagnosed after menopause, with an incidence peak during the seventh decade of life. Its symptoms arise early in the disease course, and therefore patients frequently present with early-stage disease [2]. Diagnosis at an early or advanced stage is associated with high survival rates or poor prognosis, respectively [3]. Therefore, the search for specific biomarkers for early-stage endometrial carcinoma has gained increasing attention [4]. Furthermore, patients with a more aggressive histopathology reportedly account for the majority of individuals presenting with stage III or IV disease that are typically non-responsive to gold standard chemotherapy treatment involving carboplatin plus paclitaxel, and have a poor prognosis. The identification of novel treatment modalities for patients diagnosed with biologically aggressive endometrial carcinoma remains an unmet medical need [5]. In recent years, the role of Bmi-1 in tumor cell escape from apoptotic cell death and in the failure of chemotherapy has been identified [6,7]. Bmi-1 is critically implicated in carcinogenesis and chemoresistance, and the modulation of its expression has emerged as a therapeutic strategy in the endometrial cancer field [8].

Bmi-1 is a member of the polycomb group genes, which are important transcriptional regulators through the formation of polycomb group bodies and chromatin remodeling [9]. Furthermore, it plays a significant role in cell cycle regulation, cell immortalization, and cell senescence. Numerous studies have demonstrated that Bmi-1 is involved in the regulation of self-renewal and differentiation of stem cells [10,11]. So far, overexpression of Bmi-1 has been reported in association with human carcinomas such as hematologic cancer [7], pancreatic cancer [12,13], cervical cancer [14], laryngeal squamous cell carcinoma [15], and gastric cancer [16,17]. Several reports have identified Bmi-1 overexpression in endometrial carcinoma. Bmi-1 levels are significantly elevated in both type I and type II models of endometrial cancer cell lines [18]. Furthermore, Bmi-1 expression is strongly associated with the enhanced invasive activity of endometrial carcinoma cell lines [19]. Engelsen et al. conducted a tissue microarray, which showed strong Bmi-1 nuclear staining in tumor cells [20]. Honig et al. demonstrated significant Bmi-1 upregulation in human breast, ovarian, endometrial, and cervical cancer specimens as compared to that in benign controls [21]. 

Bmi-1 may represent a promising target for the prevention and therapy of various cancer types [22]. In endometrial carcinoma, a second-generation inhibitor of Bmi-1 (PTC-028) decreased the invasion of endometrial cancer cells and potentiated caspase-dependent apoptosis. Therefore, anti-Bmi-1 strategies may represent a promising targeted approach in patients with advanced or recurrent endometrial cancer, a population where treatment is very challenging [18]. However, to date, there has been limited research into the role of Bmi-1 in endometrial cancer. Only a few studies have directly addressed the relationship between Bmi-1 expression and its significance as a clinicopathological parameter, and Bmi-1 has not yet been established as a proliferation or prognostic marker. 

Accordingly, the aim of the present study was to clarify the value of Bmi-1 as a marker of cell proliferation and its prognostic ability in endometrioid carcinoma. For this purpose, we performed immunohistochemical staining of endometrial carcinoma and normal endometrial tissue. Recently, anti-Bmi-1 treatment has emerged as a promising strategy for endometrial carcinoma. Thus, our study provides fundamental data regarding the anti-Bmi-1 strategy.

## 2. Materials and Methods

### 2.1. Tissue Samples

The present study was approved by the Ethics Committee of the Hirosaki University School of Medicine (2013-232). Informed consent was obtained from all patients prior to the beginning of the study. Untreated (such as chemotherapy and hormone therapy) surgical samples were collected at the Hirosaki University Hospital from 48 patients with endometrial carcinomas, including 16, 15, and 17 patients with grade 1 (G1), grade 2 (G2), and grade 3 (G3) endometrioid adenocarcinoma, respectively. Normal endometrial tissue samples (4 in the proliferative phase and 3 in the secretory phase) were collected from patients with benign gynecological diseases. The surgical staging and histological diagnosis were performed based on the criteria established by the International Federation of Gynecology and Obstetrics (FIGO) Staging System 2008 [23,24], including 18, 9, 8, 11, and 2 patients with Stage I A, I B, II, III, and IV.

### 2.2. Immunohistochemical Staining

Blocks of normal and endometrial carcinoma tissue were routinely processed for 10% formalin-fixing and paraffin-embedding. Next, 3-μm-thick serial sections were cut and placed on glass slides for immunohistochemical staining. Bmi-1 and Ki-67 staining was performed using a Ventana Discovery XT (Roche Diagnostics KK, Tokyo, Japan). Briefly, sections were incubated in cell conditioning solution CC1 (Roche Diagnostics KK), then incubated in peroxidase-blocking solution (Inhibitor D, Roche Diagnostics KK), and heated at 100 °C for 60 min. The following primary antibodies were applied: anti-human Bmi-1 (rabbit monoclonal antibody, clone D20B7, 1:100, Cell Signaling Technology KK, Tokyo, Japan) and anti-human Ki-67 (mouse monoclonal antibody, clone MIB-1, 1:50, DakoCytomation A/S, Glostrup, Denmark). After incubation for 32 min at 37 °C, the tissue sections were incubated with a universal secondary antibody (Roche Diagnostics KK) for 20 min at 37 °C and then visualized by the DAB Map detection kit (Roche Diagnostics KK). Nuclei were counterstained using a hematoxylin counterstain reagent (Roche Diagnostics KK). Cyclin A and p53 were visualized utilizing an EnVision^TM^ detection system (DakoCytomation A/S) following the manufacturer’s recommendations. We performed an antigen retrieval step in 10 mM citrate buffer (pH 6.0) before the immunohistochemical staining. Sections for cyclin A were boiled in a microwave oven for 20 min, whereas autoclave processing for 20 min at 121 °C was used for p53. Endogenous peroxidases in the specimens were blocked with the peroxidase-blocking solution (DakoCytomation A/S) for 5 min at room temperature, followed by incubation with a protein block serum-free reagent (DakoCytomation A/S). Sections were incubated with the following primary antibodies: anti-mouse cyclin A2 monoclonal antibody (clone 6E6, 1:15; Abcam) for 30 min at room temperature or anti-mouse p53 monoclonal antibody (clone DO-7, 1:80; DakoCytomation A/S) for 60 min at room temperature. Subsequently, the tissue sections were incubated with EnVision^TM^ /HRP rabbit/mouse secondary antibodies (DakoCytomation A/S) for 30 min at room temperature and then with the chromogen 3,3′-diaminobenzidine. Nuclei were counterstained using Mayer’s hematoxylin (Wako Pure Chemical Industries Ltd.). The specimens were examined and photo-graphed at ×200 magnification utilizing a digital microscope camera (Olympus AX80 DP21; Olympus, Tokyo, Japan) interfaced with a computer. All protein levels were evaluated using the nuclear labeling index (%), recorded as the percentage of positively stained nuclei in 100 cells in the hot spot.

### 2.3. Statistical Analysis

All statistical tests were performed using BellCurve for Excel ver. 3.10 software (Social Survey Research Information, Tokyo, Japan) and Kruskal-Wallis analysis with the Steel-Dwass post-hoc test. *p* < 0.05 was considered to indicate statistical significance. Data are presented as mean percentages of positive cells ± standard deviation. The correlation between the nuclear labeling indexes of Bmi-1 and Ki-67, cyclin A, or p53 was assessed by Spearman’s rank correlation.

## 3. Results

### 3.1. Bmi-1 Protein Expression in Normal and Cancer Endometrial Tissues Determined by Immunohistochemical Staining

We determined the immunohistochemical staining of Bmi-1, which is expressed mainly in the nucleus (Figure 1F). Bmi-1 levels in endometrial carcinoma G2 (Figure 1D) and G3 (Figure 1E) were higher than those in normal endometrial tissues in the secretory (Figure 1A) and proliferative phases (Figure 1B). 

Next, we determined the nuclear labeling index of Bmi-1 (Figure 2A). It was significantly higher in all grades of endometrial carcinoma than in secretory phase normal endometrium tissue. Similarly, the Bmi-1 nuclear labeling index was significantly higher in G2 and G3 endometrial carcinoma than in proliferative phase tissue. However, it did not differ between secretory and proliferative phase normal endometrial tissue. Furthermore, we found no significant differences in the Bmi-1 nuclear labeling index in tissues with different FIGO stages (Figure 2B). 

### 3.2. The Nuclear Labeling Index of Bmi-1 and Ki-67, Cyclin A, and p53 in Endometrial Carcinoma

Furthermore, we compared the nuclear labeling index of Bmi-1 and other proliferative and prognostic markers (Ki-67, cyclin A, and p53 proteins) using semi-serial sections (Figure 3). Both Bmi-1 and Ki-67 were more strongly expressed in carcinoma than in normal endometrial tissues (Figure 3A). Bmi-1 levels were significantly higher in carcinoma (G1–G3) than in the secretory phase; similarly, Ki-67 expression was significantly higher in G3 than in the secretory phase. In endometrial carcinoma, Ki-67 expression was significantly higher in G3 than in G1. Furthermore, Bmi-1 showed a tendency toward an increase in G2 and G3 in comparison to G1 tissues. Bmi-1 and cyclin A expression showed a tendency to increase in endometrial carcinoma in comparison to normal tissues. However, both proteins showed no significant differences (Figure 3B). P53 was expressed in endometrial carcinoma but not in normal endometrial tissue. Furthermore, it was significantly higher in G2 and G3 than in G1. Likewise, Bmi-1 had a tendency to show a higher expression in G2 and G3 than in G1 (Figure 3C).

### 3.3. Correlation of Bmi-1 Expression with Proliferation and Prognostic Markers in Endometrial Carcinoma 

The nuclear labeling index of Bmi-1 in endometrial carcinoma was positively correlated with those of Ki-67, cyclin A, and p53 using Spearman’s rank correlation test (Table 1). A highly significant correlation between Bmi-1 and Ki-67 (rs = 0.9; *p* = 0.037), cyclin A (rs = 0.9; *p* = 0.037), and p53 (rs = 0.97; *p* = 0.004) was observed.

## 4. Discussion

Overexpression of Bmi-1 has been detected in many human cancer types, including endometrial carcinoma [21]. We examined Bmi-1 expression in endometrial carcinoma tissue and normal tissue (in the secretory and proliferative phases) using immunohistochemical staining. Furthermore, we determined that Bmi-1 overexpression correlates with other proliferation prognostic markers (Ki-67, cyclin A, and p53 proteins). In our preliminary experiments using Ishikawa and HEC-50B derived from patients with low-grade and high-grade endometrial carcinoma cell lines, Bmi-1 was highly expressed in both cell lines (data not shown). Consistent with our results, Buechel M et al. reported that Bmi-1 levels were higher in both type I and type II models of endometrial carcinoma cell lines than in normal endometrial cells [18]. Bmi-1 expression determined by immunohistochemistry was significantly higher in endometrial carcinoma than in normal endometrial tissues, which is in agreement with previous studies [18,20]. 

Furthermore, we tested whether Bmi-1 expression advances in different grades of endometrial carcinoma. There was a tendency for higher Bmi-1 expression in G2 and G3 than in G1, but it was not significant (Figure 2A). These results were similar to the results observed in previous studies that showed no significant associations between Bmi-1 expression and histological grade [20]. These findings were also in agreement with those from our previous study on Ishikawa (G1) and HEC 50B (G3) cell lines using immunocytochemical staining.

In human breast cancer specimens, Bmi-1 expression was most pronounced in the invasive front of the tumor [21], but in our study of endometrial carcinoma, diffuse Bmi-1 expression was observed, and no invasive front could be confirmed.

Bmi-1 overexpression in ovarian cancer is reportedly strongly correlated with histological grade and clinical stage [25]. Furthermore, it potentiates metastasis via the induction of epithelial-mesenchymal transition in endometrial carcinoma [19]. However, another report found no significant association between Bmi-1 expression and changes in the FIGO stage [20]. Likewise, our data did not detect significant differences between Bmi-1 expression and clinical stage (Figure 2B). Therefore, even if detection is delayed in cases with low proliferative potential and low malignancy, lymph node or distant metastasis would allow detection. Thus, the Bmi-1 indication of malignancy and stage are not always correlated. However, an association between Bmi-1 and prognosis has been reported, and the relationship between them may need to be investigated on a larger number of cases and with different evaluation methods.

The expression of both Bmi-1 and Ki-67 was higher in carcinoma than in normal endometrial tissue. Furthermore, Ki67 expression was significantly higher in G3 than in G1, whereas Bmi-1 had a tendency to increase in G2 and G3 in comparison to G1 (Figure 3A). Ki-67 is recognized as an independent prognostic factor [26] and a potential prognostic biomarker for endometrial carcinoma [27,28]. Furthermore, Ki-67 is widely accepted as a cell proliferation marker [29,30].

Bmi-1 exhibited a similar expression pattern to cyclin A, and both proteins showed a tendency towards upregulation in endometrial carcinoma in comparison to normal tissues (Figure 3B). Moreover, Bmi-1 and cyclin A were positively correlated as determined by Spearman’s rank correlation test (Table 1). These results suggested that Bmi-1 has a similar proliferation potential to cyclin A. In our previous study, cyclin A expression was significantly increased in cells with higher proliferative ability; furthermore, cyclin A was specifically expressed in cells that had passed the G1/S checkpoint, whereas Ki-67 expression was not associated with changes in the cell cycle or established variations in differentiation between cell lines. Cyclin A was apparently a more accurate indicator of elevated proliferation ability in endometrial cancer cell lines than Ki-67 [31]. 

In this study, high expression of Bmi-1 and especially p53 was observed in late-stage tumors (Figure 3C). In endometrial adenocarcinoma, p53 overexpression has been reported as a poor prognostic factor [32,33]. Furthermore, we showed a significant positive relationship between Bmi-1 and Ki-67, cyclin A, or p53 by Spearman’s rank correlation test (Table 1). Ki-67, cyclin A, and p53 protein expressions have shown an independent prognostic impact in endometrial carcinoma [34,35,36]. Our results suggest that high Bmi-1 levels can be an independent prognostic marker in endometrial carcinoma. Furthermore, in recent years, attention has been focused on individualized cancer treatment that reflects the biological characteristics of individual cases. Therefore, neoadjuvant chemotherapy or immunological and genetic evaluation before treatment has been studied intensively [37]. Molecular targeted therapy using Bmi-1 is also being evaluated for endometrial carcinoma. Thus, there is a possibility that new cancer treatment may potentially target Bmi-1 elevation. Zhai H et al. reported that the presence of miR-194, which inhibits Bmi-1 levels, was inversely correlated with tumor stage in type I endometrial cancer samples [38]. Dong et al. reported that miR-194 suppressed Bmi-1 expression, affecting epithelial to mesenchymal transition [19]. In the study by Kim et al., Bmi-1 silencing demonstrated the potential to overcome chemoresistance in endometrial cancer [8]. Buechel M et al. reported PTC-2 as a first-generation inhibitor of Bmi-1, which may represent a promising targeted approach for patients with advanced or recurrent endometrial cancer by utilizing anti-Bmi-1 strategies [18]. Patients with a higher Bmi-1 score would be expected to respond favorably to anti-Bmi-1 therapy. Thus, new endometrial cancer treatment strategies may potentially target elevated Bmi-1.

## 5. Conclusions

Our study suggested the usefulness of Bmi-1 expression as a prognostic marker in endometrial carcinoma tissues. Bmi-1 levels were significantly higher in cancer than in normal endometrial tissue. Furthermore, the nuclear labeling index of Bmi-1 in endometrial carcinoma was strongly positively correlated with that of Ki-67, cyclin A, and p53, which are well known as independent prognostic factors. Moreover, it was correlated with cyclin A, which was previously reported as a proliferation marker. Therefore, Bmi-1 may be a reliable proliferation and prognostic biomarker for endometrioid carcinoma. Recently, a Bmi-1 inhibitor showed promising results, and it might be a possible companion diagnostic to anticancer drugs in the future.

## Figures and Tables

**Figure 1 medicina-56-00072-f001:**
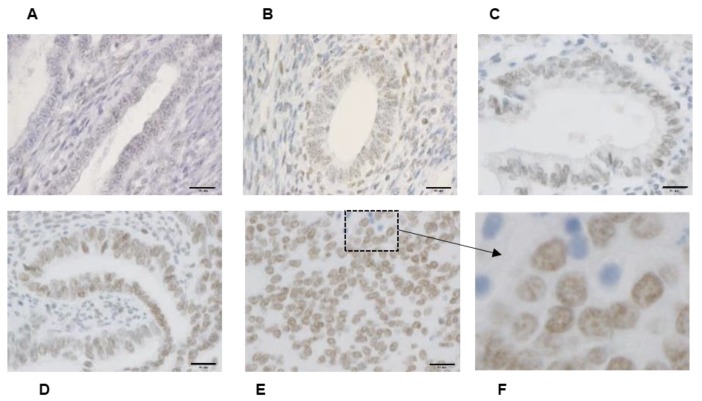
Bmi-1 immunohistochemical staining. Normal endometrial tissue includes secretory phase (**A**) and proliferative phase (**B**) endometrium. Different grades of histodifferentiation in endometrial cancer. G1: Well-differentiated (**C**), G2: moderately differentiated (**D**), and G3: poorly differentiated (**E**). (**F**) is enlarged for clarity of the boxed area shown in panel (**E**). (**A**–**E**): Scale bar = 20 μm.

**Figure 2 medicina-56-00072-f002:**
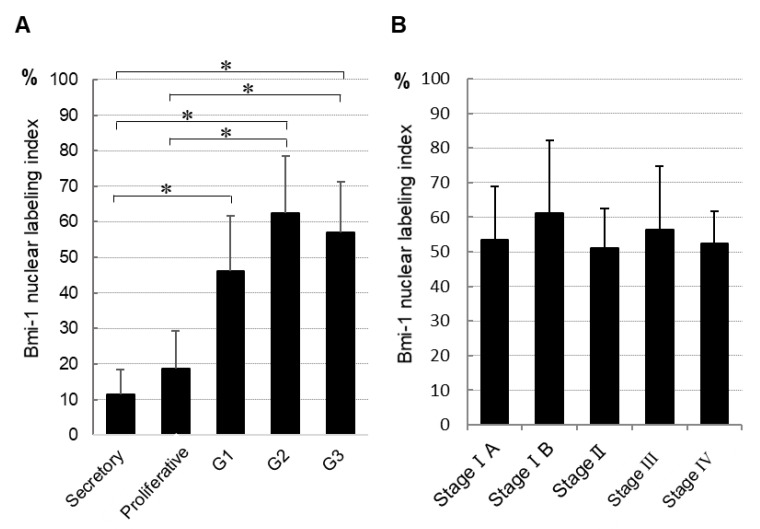
Bmi-1 nuclear labeling index of normal tissue and each histological grade of endometrial cancer tissues (**A**) and different endometrial cancer stages (**B**). The data shown are means of nuclear labeling index ± standard deviation; * *p* < 0.05.

**Figure 3 medicina-56-00072-f003:**
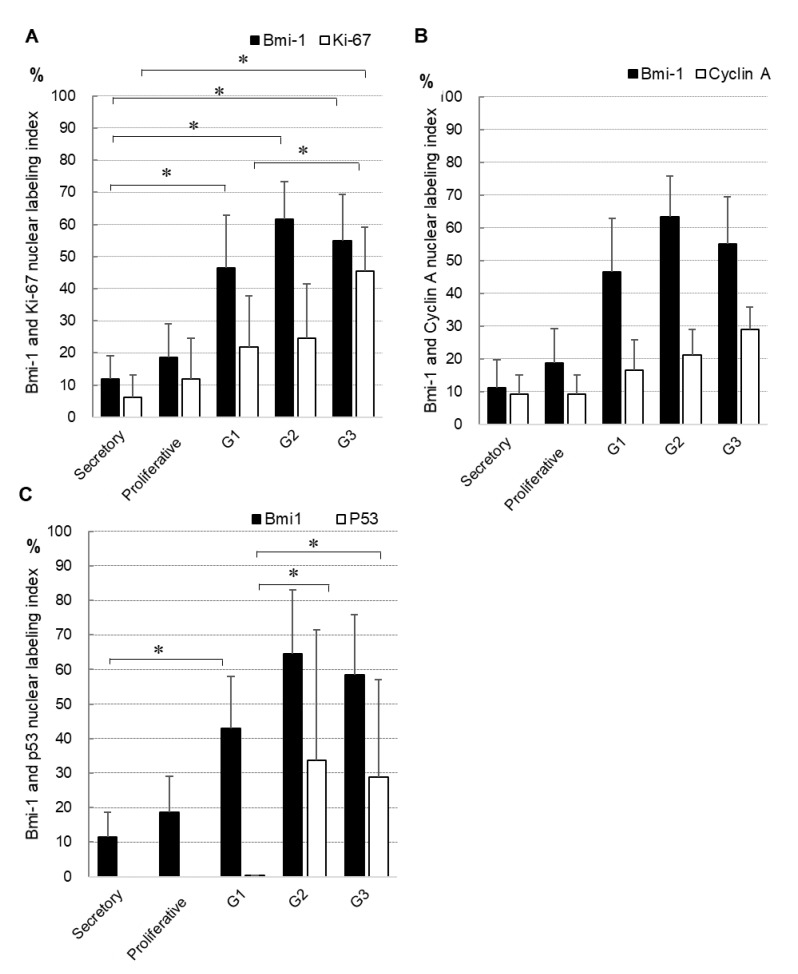
Comparison of the nuclear labeling index of Bmi-1 and Ki67 (**A**), cyclin A (**B**), or p53 (**C**) in semi-serial sections. Data shown are means of nuclear labeling index ± standard deviation; * *p* < 0.05.

**Table 1 medicina-56-00072-t001:** The correlation of the nuclear labeling index between Bmi-1 and Ki-67, cyclin A, or p53 was determined using Spearman’s rank correlation. * *p* < 0.05, ** *p* < 0.01.

Histological Type	Correlation (rs)	*p*-Value	Number of Cases
Bmi-1 and Ki-67	0.9	0.037 *****	37
Bmi-1 and cyclin A	0.9	0.037 *****	35
Bmi-1 and p53	0.97	0.004 ******	23

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
