# Peer review of "Bmi-1 Immunohistochemical Expression in Endometrial Carcinoma is Correlated with Prognostic Activity"

_medicina, 2020, doi:10.3390/medicina56020072_

Round 1

Reviewer 1 Report

The presented paper reports on the IHC expression  of Bmi-1 protein in endometrial cancer in relation with other known histological and IHC markers.

Although the work is interesting and well exposed, it is not really innovative, since hundreds of novel prognostic factors are proposed each year.

Some minor concerns follow:

Please provide a summary table on the general characteristics of the study population, divided in Cases Vs Controls, at the beginning of the result section (E.g. Numerosity, Age, FIGO, Parity, Body Mass Index... etc) The staining pattern of non-neoplastic endometrial stroma should be elucitated The paper title could be misleading since nor overall survival (OS) neither disease free survival (DFS) are covered in the text  

Reviewer 2 Report

I read with great interest the work entitled "Bmi-1 immunohistochemical expression in endometrial carcinoma is correlated with prognostic activity". The authors emphasize the need to look for new solutions for the treatment of endometrial cancer, indicating the expression of Bmi-1 as a useful prognostic marker. In my opinion, the topic is of interest, however minor corrections should be made.

I recommend revising the main text with the following points:

- The authors used samples from 48 patients diagnosed with endometrial cancer and 7 normal endometrial tissue samples. What were the inclusion and exclusion criteria (e.g. the use of hormone replacement therapy)?

- The authors use staging and grading of endometrial cancer in accordance with FIGO criteria. The exact number of samples for each grade is given in the paper, however there is no such information regarding staging. The presented results refer to both grading and staging, so it would be valuable to include the number of samples for each cancer stage.

- For statistical analysis of the results, the authors used the Kruskal-Wallis and Steel-Dwass post-hoc tests, which are non-parametric. Can the authors indicate why they presented the results as mean and standard deviation and did not use the median?

- In the caption under Figure 1, I recommend to replace the sentence "G1: Well differentiation degree (C), G2: moderate differentiation degree (D), and G3: poor differentiation degree (E)." with "G1: well-differentiated (C), G2: moderately differentiated (D), and G3: poorly differentiated (E)."
